# Germline Mutations in Other Homologous Recombination Repair-Related Genes Than *BRCA1/2*: Predictive or Prognostic Factors?

**DOI:** 10.3390/jpm11040245

**Published:** 2021-03-28

**Authors:** Laura Cortesi, Claudia Piombino, Angela Toss

**Affiliations:** Genetic Oncology Unit, Department of Oncology and Haematology, University Hospital of Modena, 41125 Modena, Italy; claudia.piombino@outlook.com (C.P.); angela.toss@unimore.it (A.T.)

**Keywords:** BRCA1, BRCA2, PALB2, homologous recombination repair

## Abstract

The homologous recombination repair (HRR) pathway repairs double-strand DNA breaks, mostly by BRCA1 and BRCA2, although other proteins such as ATM, CHEK2, and PALB2 are also involved. *BRCA1/2* germline mutations are targeted by PARP inhibitors. The aim of this commentary is to explore whether germline mutations in HRR-related genes other than *BRCA1/2* have to be considered as prognostic factors or predictive to therapies by discussing the results of two articles published in December 2020. The TBCRC 048 trial published by Tung et al. showed an impressive objective response rate to olaparib in metastatic breast cancer patients with germline *PALB2* mutation compared to germline *ATM* and *CHEK2* mutation carriers. Additionally, Yadav et al. observed a significantly longer overall survival in pancreatic adenocarcinoma patients with germline HRR mutations compared to non-carriers. In our opinion, assuming that *PALB2* is a high-penetrant gene with a key role in the HRR system, *PALB2* mutations are predictive factors for response to treatment. Moreover, germline mutations in the *ATM* gene provide a better outcome in pancreatic adenocarcinoma, being more often associated to wild-type *KRAS*. In conclusion, sequencing of HRR-related genes other than *BRCA1/2* should be routinely offered as part of a biological characterization of pancreatic and breast cancers.

## 1. Introduction

### 1.1. DNA Repair Mechanisms

DNA damage and deficiencies of repair are central features of cancer pathology. Healthy cells defend themselves against DNA damage through different pathways. The DNA damage can induce a single-strand break or a double-strand break. The single-strand break can be repaired by different systems: base excision repair (BER), nucleotide excision repair (NER), or mismatch repair (MMR). The poly (ADP-ribose) polymerase (PARP) enzyme belongs to the BER system (Figure 1A), whereas ERCC excision repair 1, endonuclease non-catalytic subunit (ERCC1) enzyme repairs bulky adducts by the NER systems, and the MutL homolog 1 (MLH1), MutS homolog 6 (MSH6), and MutT homolog 1 (MTH1) proteins repair the single nucleotide substitutions, deletions, or insertions by the MMR system. In eukaryotic cells, there are at least five pathways to repair DNA double-strand breaks: non-homologous end-joining (NHEJ), alternative non-homologous end-joining, single-strand annealing, break-induced replication, and homologous recombination repair (HRR) [1,2,3].

The most accurate of all, HRR, uses the intact sister chromatid as a template for error-free DNA double-strand break repair, mainly during the S/G2 phase of the cell cycle. DNA damage response is fundamentally mediated by the kinases belonging to the phosphatidylinositol 3-kinase-like protein kinase family, which include ataxia telangiectasia mutated (ATM), ataxia telangiectasia and Rad3-related (ATR), and DNA-PK (DNA-dependent protein kinase). While DNA-PK activates the more error-prone template-independent mechanism of NHEJ [4], both ATM and ATR orchestrate the initial phase of the HRR pathway and mediate cell cycle arrest [5]. In detail, the MRN complex (Mre11, Rad50, and Nbs1) initiates DNA end resection from 5′ to 3′ leading to the formation of single-strand DNA (ssDNA) at the extremity of the DNA double-strand break repair. ssDNA is protected from degradation by the loading of replication protein A (RPA) [6]. The MRN complex recruits and activates ATM [7], while the sensor protein RPA finally drives ATR activation [8]. Once activated, ATM and ATR phosphorylate several proteins involved in the HRR pathway such as checkpoint kinase 2 (CHEK2) [5]. On the other hand, the tumor suppressor complex Breast Cancer 1 (BRCA1) and (BRCA1-associated RING domain 1 (BARD1) facilitates DNA end resection and interacts with the bridging protein partner and localizer of BRCA2 (PALB2) which in turn promotes the recruitment of breast cancer 2 (BRCA2) [9]. PALB2 and BRCA2 remove RPA and facilitate the assembly of the RAD51 recombinase nucleoprotein filament. RAD51 nucleoprotein filament mediates the invasion of ssDNA into the intact sister chromatid, searching for a homologous template for DNA synthesis and faithful repair of DNA [10] (Figure 1C,D).

### 1.2. PARP Inhibitor Treatments

We well know that tumor cells with *BRCA1/2* germline mutations are targeted by PARP inhibitor (PARPi) therapies through synthetic lethality. During DNA replication, PARPi induces the single arm of the fork interruption, producing a collapsed fork (Figure 1B). If PARP enzymes are inhibited in cells lacking functional BRCA1/2 proteins, DNA double-strand breaks can only be repaired by the NHEJ pathway. However, the error-prone nature of this template-independent repair pathway ultimately leads to tumor cell death [11]. Since 2009, when a first-in-human clinical trial of olaparib confirmed the synthetic lethal interaction between inhibition of PARP1, a key sensor of DNA damage, and *BRCA1/2* deficiency [12], PARPi therapies have been approved for the use in several cancers. Based on Study 19 [13], Study 42 [14], and SOLO2 studies [15], olaparib has been approved for the response maintenance treatment of germline/somatic *BRCA1/2*-mutated high-grade serous ovarian cancers (including fallopian tube or primary peritoneal cancers) after first-line platinum-based chemotherapy and for the treatment of germline *BRCA1/2* (g*BRCA1/2*) mutated ovarian cancer progressing to three or more prior lines of chemotherapy. Besides, another two PARP inhibitors, niraparib [16], and rucaparib [17] have been granted approval in the maintenance setting of ovarian cancer, regardless of *BRCA1/2* status. Furthermore, both olaparib [18] and talazoparib [19] are approved in human epidermal growth factor receptor 2 (HER2)-negative, g*BRCA1/2*-mutation-associated metastatic breast cancer [20]. Finally, several trials of olaparib in patients with g*BRCA1/2* mutations identified responders beyond ovarian or breast cancer patients, suggesting that other HRR-defective tumors could be suitable for PARPi treatment [14].

### 1.3. PARPi Targeting HRR Genes other Than BRCA

Regarding breast cancer, in addition to the known high-penetrance pathogenic variants of *BRCA1/2*, mutations in other high- or intermediate-penetrance genes can increase the risk of cancer [21] and the most common non-*BRCA* pathogenic or likely pathogenic variants affect *PALB2*, *ATM*, and *CHEK2* genes [22]. Individuals carrying heterozygous pathogenic variants of *ATM* have a 33% cumulative lifetime risk of breast cancer by 80 years of age [23]. Nevertheless, *ATM* heterozygous pathogenic variants have also been reported in some cases of familial ovarian, pancreatic, and prostate cancer [21]. Certain variants in *CHEK2* are associated with increased breast cancer risk, with a cumulative lifetime risk ranging from 28% to 37%, depending on family history [24]. Within families carrying pathogenic CHEK2 variants, there is also an increased risk of other malignancies including colon, prostate, kidney, bladder, and thyroid cancers [25].

With the aim of testing the hypothesis that olaparib would also have efficacy in germline mutation in an HRR-related gene other than *BRCA1/2,* the TBCRC-048 study was designed in metastatic breast cancer patients. In this study, eligible patients had germline mutations in non-*BRCA1/2* HRR-related genes (cohort 1) or somatic mutations in these genes or *BRCA1/2* (cohort 2). Fifty-four patients received olaparib 300 mg orally twice a day until progression. Exclusion criteria included platinum refractory disease or progression on more than two chemotherapy regimens in the metastatic setting. Examining cohort 1, this phase II trial found that the olaparib provided an impressive objective response rate (82%) in patients with germline *PALB2* (g*PALB2)* mutation compared to germline *ATM* (g*ATM*) and germline *CHEK2* (g*CHEK2)* mutation carriers [26]. Therefore, olaparib could be used in g*PALB2* mutation carriers beyond patients with g*BRCA1/2* mutations, significantly expanding the number of patients with metastatic breast cancer who would benefit from PARPi. Moreover, since g*PALB2* mutations also predispose to pancreatic and ovarian cancers [27,28], these results may have significant implications for the treatment of other g*PALB2*-associated cancers. Reasons why g*PALB2* mutation carriers are so highly responsive to olaparib compared to g*ATM* and g*CHEK2* have not been fully explained. Nonetheless, similar data have been found in a phase II trial that evaluated talazoparib in patients with advanced HER2-negative breast cancer or other solid tumors with a germline or somatic alteration in HRR-related genes other than *BRCA1/2*. Patients who received at least one prior therapy in the advanced setting and without progression on or within 8 weeks of their last platinum dose were eligible. They were treated with talazoparib 1 mg daily until disease progression. Twenty patients were enrolled: 13 breast cancers (12 luminal and 1 triple negative) and 7 other solid cancers (pancreas, colon, uterine, testicular, and parotid salivary). Of 12 breast cancer patients evaluated, 6 showed a response or stable disease (clinical benefit rate equal to 50%), of which 3 were g*PALB2* mutation carriers. No responses were observed in non-breast tumors. This proof-of-concept phase II study with talazoparib as the single agent demonstrated activity in HER2-negative advanced breast cancer patients with an HR pathway mutation beyond *BRCA1/2* [29].

The study by Yadav et al. [30] evaluated the clinical characteristics of pancreatic ductal adenocarcinoma in germline mutation carriers of HRR genes and the implications of these mutations on overall survival (OS) by analyzing 37 cancer predisposition genes in 3078 patients. One hundred seventy-five HRR mutation carriers and 2730 noncarriers were compared, finding a younger age and more metastatic disease at diagnosis in HRR mutation carriers. In a multivariable model adjusting for sex, age at diagnosis, and tumor staging, patients with germline HRR mutations had a significantly longer OS compared with noncarriers (HR, 0.83; 95% confidence interval (CI), 0.70–0.97; *p* = 0.02). Further, gene-level analysis demonstrated that germline *ATM* mutation carriers had longer OS compared with patients without germline mutations in any of the 37 genes (HR, 0.72; 95% CI, 0.55–0.94; *p* = 0.01).

The primary aim of our commentary was to discuss whether germline mutations in an HRR-related gene other than *BRCA1/2* have to be considered as prognostic factors or predictive to therapies by discussing the results of the abovementioned TBCRC-048 study by Tung et al. [25] and of the study by Yadav et al. [30].

## 2. The Different Role of High- and Moderate-Penetrance Genes

The role of *PALB2* as a high/moderate-penetrance gene has been extensively discussed. The risk of breast cancer development in g*PALB2* mutations is lower than in g*BRCA1/2* mutation carriers, reaching 53% at 80 years [31]. However, breast cancer risk appears higher than in g*ATM* or g*CHEK2* mutation carriers, where it is 2/3-fold higher than in the general population [23,24]. The difference between high- and moderate-risk genes is that in the latter case, both endogenous (genomic variations) and exogenous (e.g., environmental exposures and lifestyle) factors contribute to cancer development. It is likely that in cases of cancer in g*ATM* and g*CHEK2* mutation carriers, olaparib needs to be supported by the modification of environmental factors such as diet or lifestyle in order to improve its efficacy. It might be of interest to evaluate the addition of a methionine-choline-deficient diet to PARPi treatment. In a murine model of non-alcoholic fatty liver disease, a methionine- and choline-deficient diet attenuated PARP activation, enhancing the benefits of olaparib [32].

As previously described, *PALB2* represents a key gene in the HRR system, as the signal mediator between BRCA1 and the BRCA2/RAD51 complex. PALB2 purification studies [33,34] revealed that PALB2 uses two DNA-binding domains to interact directly with D-loop and ssDNA structures, and that the recombinase RAD51 interacts with PALB2 through its amino-terminal region, which leads to the enhancement of RAD51 activity. In its carboxyl-terminal region, PALB2 presents a WD40 domain through which PALB2 interacts with both BRCA2 and RAD51 [34,35]. Being an important player in different steps of HRR, PALB2 is strictly regulated by ubiquitylation and histone acetylation [36], as well as at a post-transcriptional level, being phosphorylated initially by cyclin-dependent kinases (CDKs) [37] and later by ATM [38]. As in *BRCA1/2* mutation carriers, the lack of both *PALB2* alleles causes the activation of the NHEJ with a consequent genomic instability and cancer cell death [11].

On the other hand, ATM has multiple functions in cancer development, such as cell cycle checkpoint modulation, DNA double-strand break repair, metabolic regulation, migration, and chromatin remodeling [39]. Following exposure to stress, ATM acts as a cell cycle checkpoint regulating G1/S arrest, S phase, and G2-M arrest after the DNA double-strand break repair through different pathways. CHEK2 is also involved in the cell cycle arrest, being activated by ATM [40]. In case of g*ATM* or g*CHEK2* mutation, their role in cell cycle arrest can be overcome by ATR, which blocks the cell cycle, allowing the HRR before the cell enters into replication and mitosis phases. In detail, while ATM identifies and amplifies the signal generated by DNA double-strand break repair, ATR is activated by ssDNA or interstrand DNA crosslinking (both of which lead to stalled replication forks), or by resected DNA double-strand break repair. The principal substrates of ATM and ATR are the checkpoint kinases CHEK2 and CHEK1, respectively, which block cell cycle progression to allow repair. In particular, CHEK2 triggers the G1-S checkpoint by phosphorylation of the tumor-suppressor protein p53 which in turn inhibits the cyclin-dependent kinase 2 (CDK2)-CCNE1 (cyclin E1) interaction. In contrast, CHEK1 is mainly involved in the intra-S checkpoint and the G2-M checkpoint [41]. It is therefore likely that in both germline mutations, the PARPi needs to be added to ATR or CHEK1 inhibitors to work more effectively than in the case of g*PALB2* mutations. Besides, the ATR–CHEK1 pathway is often upregulated in human neoplasms, especially CHEK1, whose promoter activity is believed to promote tumor growth. Nevertheless, some evidence indicates that ATR and CHEK1 may also behave as haploinsufficient oncosuppressors, at least in a specific genetic background. Interestingly, the inactivation of ATM–CHEK2 and ATR–CHEK1 pathways in preclinical studies has been shown to efficiently sensitize malignant cells to radiotherapy and chemotherapy [41].

On these grounds, we can conclude that g*PALB2* mutations represent a predictive factor for treatment. However, can we also consider g*PALB2* mutations as having a prognostic role? The prognostic versus predictive nature of HRR defect deserves special attention. Recent results from the study by Yadav et al. [30] showed a significantly longer OS in patients with germline HRR (gHRR) mutations compared to non-carriers in pancreatic ductal adenocarcinoma. This is particularly evident in g*ATM* mutation carriers, although patients with gHRR mutations more often present metastatic disease at the diagnosis. Molecular studies previously showed that gHRR mutations are more frequently associated to the wild-type *KRAS* gene, which is one of the best prognostic factors in pancreatic ductal adenocarcinoma. Therefore, the pancreatic ductal adenocarcinoma with wild-type *KRAS* seems to have the most favorable prognosis when accomplished by gHRR mutations, which is probably responsible for the best response to therapy.

## 3. Results

### 3.1. How to Improve the Access to Genetic Counseling and Testing

All trials that explore the efficacy of PARPi in high-penetrance risk genes (*BRCA1/2, PALB2*) have shown activity in those genes, definitely introducing in the metastatic breast cancer therapeutic algorithm a new standard treatment for germline mutation carriers. Unfortunately, patients with breast cancer who may be eligible for PARP inhibitor therapy are often missed, even when using established diagnostic guidelines and techniques. In the USA, only 2.7% of eligible women reported the uptake of genetic counseling and testing [42,43]. Eligibility for and uptake of gHRR testing varies among countries [44,45], and the use of international testing criteria is not feasible for all countries owing to disparities in resources and ethnicities [46].

There are potential barriers to gHRR testing and genetic counseling for eligible women with or without a diagnosis of breast cancer: lack of understanding and knowledge about genetic counseling and testing by physicians and patients; lack of perceived benefits of counseling; lack of perceived risk of having a mutation; cost of testing; and fear of insurance discrimination [47,48,49,50]. Patients’ attitudes to gHRR testing (the predisposing factor), income (the enabling factor), and risk of carrying a mutation (the need factor) predict uptake of testing [51]. There are multiple ways that the uptake of gHRR testing may be increased: provision of free genetic counseling; greater dissemination of information to at-risk individuals; genetic counseling that covers strategies for individuals to discuss their diagnosis with family members; and awareness and implementation of population-based testing as a preventive measure [52,53].

Future avenues to identify patients with gHRR mutations who may benefit from treatment with PARP inhibitors are under evaluation in clinical trials, and have yet to gain approval from licensing authorities, including the FDA and EMA. Moreover, in order to increase the detection of actionable genetic mutations at earlier stages of disease, a wider access to multiple-gene panel testing and the validation of predictive models to establish probabilities of having gene mutations are needed [54,55]. The evaluation of mutations in various HRR genes could be fundamental to identify patients suitable for PARP inhibitor therapy. Accordingly, a suite of biomarkers correlating with PARP activity have recently been identified in human cancer cell lines, and these could be used as patient selection criteria for expanding the clinical development of PARP inhibitors [56,57].

Interestingly, gHRR mutations also represent a favorable factor in pancreatic ductal adenocarcinoma, being more often associated to the wild-type *KRAS* gene. Hence, sequencing of gHRR genes other than *BRCA1/2* should be routinely offered as part of the biological characterization of pancreatic ductal adenocarcinoma. For all of these reasons, there is an urgent need for new techniques to aid in diagnosis, staging, and clinical-therapeutic decisions. Finally, since gastroenterology providers interface with patients who develop pancreatic ductal adenocarcinoma, they should have an understanding of genetic counseling and should be able to interpret multigene panel test results.

### 3.2. How to Improve PARPi Response in gHRR Mutation Carriers

Many efforts need to be made in gHRR mutations other than g*BRCA1/2*, for example by preventing breast cancer development with lifestyle modifications or by treating genetic tumors adding other DNA damage repair gene inhibitors to PARPi. Several recent studies have indicated the potential involvement of PARP activity in promoting metabolic dysfunction [58,59]. Interestingly, various metabolic disorders have been associated with elevated oxidative stress and DNA damage, which can subsequently induce PARP activity, which is also true in some cancers [32]. A methionine- and choline-deficient diet could increase the PARP inhibitor activity in gHRR mutation carriers, avoiding the development of breast cancer. A strong relationship can be observed between genetic and behavioral risk factors, underlining the prognostic role of moderate-penetrance genes.

On the other hand, as already shown, alterations in non-*BRCA1/2* HRR genes may confer sensitivity not only to PARP inhibitors but also to the WEE1 checkpoint inhibitor or the ataxia telangiectasia and Rad3-related protein inhibitor. The goal for the use of HRR-targeted agents in cancer treatment should be to maximize DNA damage in G1 and S phase, and to prevent repair in G2 phase, in order to ensure the damage is taken through into mitosis where the effects will manifest. Patients carrying HRR gene mutations, who are unable to repair the DNA double-strand break, obtain the most efficient cure by G1/S checkpoint abrogation and G2/M checkpoint prevention. Achieving similar successes in the clinic should be possible using targeted HRR agents, but most likely in combination with other targeted therapies, and will require the correct identification of cancer-specific genetic deficiencies that will be associated with susceptibility to the specific HRR-targeted agent. In addition, in order to target the right tumors with HRR agents, it will also be important to maximize the therapeutic window by identifying the correct dose and schedule for treating patients, and this in turn will require an understanding of the drug mechanism of action, target engagement, and downstream pharmacodynamic biomarkers that can be used in the clinic.

The combination of PARP inhibitors and WEE1 or ATM inhibitors is under investigation in different studies with the aim of exploiting the replication stress in these mutation carriers. For instance, the Violette trial is an ongoing study with olaparib monotherapy versus olaparib in combination with an inhibitor of ATR (ceralasertib [AZD6738]) and olaparib monotherapy versus olaparib in combination with an inhibitor of WEE1 (adavosertib [AZD1775]), in the second- or third-line setting, in patients with triple-negative breast cancer prospectively stratified by qualifying tumor mutations in genes involved in the HRR pathway [60]. Another interesting phase I study is currently evaluating the addition of adavosertib to olaparib in refractory solid tumors [61].

Additionally, other ongoing trials are evaluating PARP inhibitors in pancreatic ductal adenocarcinoma patients with non-*BRCA* DNA damage responsive gene deficiencies, as well as PARP inhibitors in combination with other agents (i.e., immune checkpoint inhibitors) to expand the group of patients that might derive benefit from this treatment.

## 4. Conclusions

To conclude, gHRR mutations represent an interesting predictive factor for treatment with DNA damage repair agents, particularly in tumors where current standard therapies are insufficient, such as in the case of pancreatic ductal adenocarcinoma. Indeed, several DNA damage repair agents are under development in order to improve the therapeutic paraphernalia in rare cancer diseases. We believe that future research should be directed to better clarifying the biological rationale underpinning the mechanism of action of moderate- and high-penetrance risk genes. A deeper knowledge of these molecular pathways is key to better understanding and exploiting the huge potential of PARP inhibitors as therapeutic agents for a wider but targeted population of cancer patients.

## Figures and Tables

**Figure 1 jpm-11-00245-f001:**
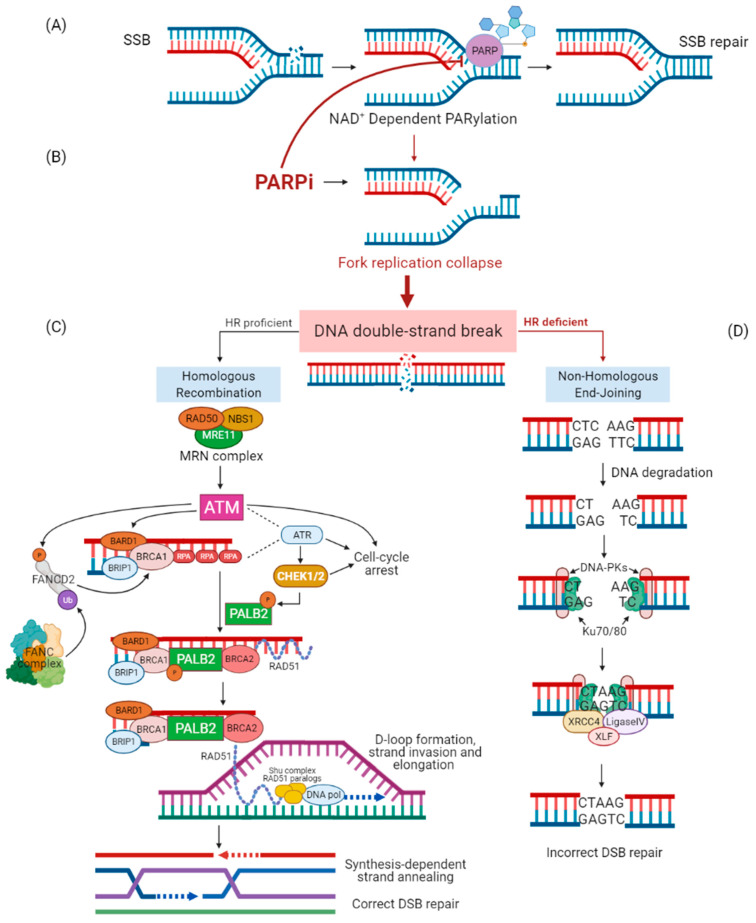
Overview of DNA double-strand break repair mechanisms and PARP inhibitor function. When DNA single-strand break (SSB) occurs, poly (ADP-ribose) polymerase (PARP) recruitment and activation leads to SSB repair through NAD+poly(ADP-ribosyl)ation (PARylation) of histones and chromatin remodeling enzymes and recruitment of PARP-dependent DNA-repair proteins (**A**). In the presence of PARP inhibitor (PARPi), PARP recruited to DNA SSB is no longer able to activate PARP-dependent repair systems and to dissociate from DNA-determining fork replication collapse during DNA replication (**B**). The collapsed replication fork creates a DNA double-strand break (DSB) that, in homologous recombination (HR)-proficient cells, is mainly repaired by the error-free mechanism of HR. MRN complex (Mre11, Rad50, and Nbs1) initiates DNA end resection, leading to the formation of single-strand DNA (ssDNA) at the extremity of the DSB; ssDNA is protected from degradation by the loading of replication protein A (RPA). The MRN complex recruits and activates ataxia telangiectasia mutated (ATM); ATM and RPA contribute to ataxia telangiectasia and Rad3-related (ATR) activation. Once activated, ATM and ATR phosphorylate several proteins involved in the HR pathway, such as checkpoint kinases 1 and 2 (CHEK1/2). Besides, ATM, ATR, and CHEK1/2 regulate cell cycle arrest after the DSB. Fanconi anemia complementation group D2 (FANCD2) contributes to breast cancer 1 (BRCA1) activation once monoubiquitinated by Fanconi anemia complementation (FANC) and phosphorylated by ATM. The complex BRCA1- BRCA1-associated RING domain 1 (BARD1) facilitates DNA end resection and interacts with the bridging protein partner and localizer of BRCA2 (PALB2) phosphorylated by CHEK2. PALB2 promotes the recruitment of breast cancer 2 (BRCA2). PALB2 and BRCA2 remove RPA and facilitate the assembly of the RAD51 recombinase nucleoprotein filament. RAD51 nucleoprotein filament, Shu complex (which consists of four proteins, Shu1, Shu2, Csm2, and Psy3), and RAD51 paralogs mediate the D-loop formation and strand invasion of ssDNA into the intact sister chromatid, searching a homologous template for DNA synthesis by DNA polymerase (DNA pol). The repaired DNA is resolved by synthesis-dependent strand annealing (**C**). In HR-deficient cells, DSB is mainly repaired by the more error-prone template-independent mechanism of non-homologous end-joining (NHEJ). DNA ends are recognized by the Ku70/80 heterodimer, which recruits DNA-dependent protein kinases (DNA-PKs). The X-ray repair cross complementing 4 (XRCC4)-DNA Ligase IV-XRCC4-like factor (XLF) ligation complex seals the break. However, DNA ends can degrade, leading to incorrect DSB repair (**D**).

## Data Availability

No new data were created or analyzed in this study. Data sharing is not applicable to this article.

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
