# Peer review of "Germline Mutations in Other Homologous Recombination Repair-Related Genes Than BRCA1/2: Predictive or Prognostic Factors?"

_jpm, 2021, doi:10.3390/jpm11040245_

Round 1

Reviewer 1 Report

Dear Laura Cortesi,

Dear Authors,

It was a pleasure reading your Commentary on 

Germline mutations in other Homologous Recombination Repair-related genes than BRCA1/2: predictive or prognostic factors? 

Thank you very much for your article.

Just one tiny issue:

In the Abstract you wrote:

 "by discussing the results of two articles 13 published on December 2020. "

published IN December 2020 would be correct.

I wish you all the best!

Best regards

Author Response

  1. R) In the Abstract you wrote:

 "by discussing the results of two articles published on December 2020. "

published IN December 2020 would be correct.

  1. A) Thanks for your comment. We have replaced ON with IN as suggested

Reviewer 2 Report

The commentary manuscript by Cortesi et al. attempts to nominate PALB2 versus BRCA1/2 as a potential prognostic marker in a comparative work. The paper is well structured but needs improvement to be publishable. 

Major points: 1) the pathways and inhibitors need to be presented in a cartoon to make them easier to understand and more catchy for readers.

2) The results need to be summarized in a list so that readers can better understand them.

Minor point: paper needs extensive proofreading and editing.

Author Response

Reviewer 2:

Major points:

R 1) the pathways and inhibitors need to be presented in a cartoon to make them easier to understand and more catchy for readers.

A 1) We have added a cartoon with the SSB and SSB repair (Fig.1A), the PARP inhibition and the fork replication collapse (Fig.1B), the subsequent DNA-double strand break repaired by the Homologous Recombination in case of HR proficient (Fig.1C) and by the Non-Homologues End-Joining in case of HR deficient system (Fig.1D)

R 2) The results need to be summarized in a list so that readers can better understand them.

A 2) The chapter 3: Results has been introduced and subdivided in two sections: 3.1 How to improve the access to genetic counseling and testing; 3.2 How to improve PARPi response in gHRR mutation carriers

Minor point:

R 3) paper needs extensive proofreading and editing.

A 3) The paper has been revised and modified by introducing three different sections in the Introduction chapter (1.1 DNA repair mechanisms; 1.2 PARP inhibitor treatments; 1.3 PARPi targeting HRR genes other than BRCA). The single strand break repair system has been added and explained in the DNA repair mechanisms section. The formation of fork replication collapse due to PARPi has been adjunted in the PARP inhibitor treatments section. One more reference has been provided (20).

The chapter “The different role of high and moderate penetrance genes” has been introduced, where the role of PALB2, ATM, CHEK2 and other genes have been elucidated, by analysing the TBCRC-048 study and the Yadav et al. study.

As already mentioned, the Results chapter has been added summarizing two principal points

Finally a “Conclusions” chapter has been inserted.

Round 2

Reviewer 2 Report

accepted in the present form.